# Validation of a Lithuanian-Language Version of the Brunel Mood Scale: The BRUMS-LTU

**DOI:** 10.3390/ijerph19084867

**Published:** 2022-04-17

**Authors:** Peter C. Terry, Albertas Skurvydas, Ausra Lisinskiene, Daiva Majauskiene, Dovile Valanciene, Sydney Cooper, Marc Lochbaum

**Affiliations:** 1Division of Research & Innovation, University of Southern Queensland, Toowoomba, QLD 4350, Australia; peter.terry@usq.edu.au; 2Institute of Educational Research, Education Academy, Vytautas Magnus University, 44248 Kaunas, Lithuania; albertas.skurvydas@vdu.lt (A.S.); ausra.lisinskiene@vdu.lt (A.L.); daiva.majauskiene@vdu.lt (D.M.); dovile.valanciene@gmail.com (D.V.); 3Department of Rehabilitation, Physical and Sports Medicine, Institute of Health Sciences, Faculty of Medicine, Vilnius University, 03101 Vilnius, Lithuania; 4Department of Kinesiology and Sport Management, Texas Tech University, Lubbock, TX 79409, USA; sydneyco@ttu.edu

**Keywords:** BRUMS-LTU, affect, emotion, mood profiling, Lithuania

## Abstract

Mood can be considered as a diffuse and global emotional state, with both valence and arousal characteristics, that is not directed towards a specific object. Investigation of moods in specific language and cultural contexts relies on the availability of appropriately validated measures. The current study involved the translation and validation of the Brunel Mood Scale (BRUMS) from English into Lithuanian. The 24-item, 6-factor scale, referred to as the BRUMS-LTU, was administered to 746 participants who were fluent in Lithuanian (*n*_men_ = 199 (26.7%), *n*_women_ = 547 (73.3%); age range = 17–78 years, *M* = 41.8 years, *SD* = 11.4 years). Confirmatory factor analysis showed an adequate fit of the hypothesized measurement model to the data (CFI  =  0.954, TLI  = 0 .944, RMSEA  = 0 .060 [CI 0.056, 0.064], SRMR = 0.070) and multi-sample analysis supported configural, metric, scalar, and residual invariance across genders. Concurrent measures (i.e., Perceived Stress Scale, Big Five Personality Test) correlated with subscale scores in line with theoretical predictions, supporting both convergent and divergent validity. Internal consistency coefficients of the six subscales were satisfactory. Mood scores varied significantly by gender, with men generally reporting more positive moods than women. Findings support the adequacy of the psychometric properties of the BRUMS-LTU. Thus, the scale can be recommended for use in further psychological studies of mood in Lithuania and may also be useful for applied practitioners.

## 1. Introduction

Moods are pervasive to human functioning and deeply influence how individuals interact with the world around them. A mood has been defined as “a set of feelings, ephemeral in nature, varying in intensity and duration, and usually involving more than one emotion” [1]. Moods are conceptualized as having a valence dimension that varies from positive (e.g., happy) to negative (e.g., depressed) and an arousal dimension that varies from activation (e.g., alert) to deactivation (e.g., tired) [2,3]. Moods differ from emotions in several ways; for example, compared to emotions, moods are more diffuse, of lesser intensity, longer duration, and not related to a specific cause [4,5]. Although moods often fluctuate from day to day, persistent and/or extreme negative moods represent increased risk of mental health disorders [6]. The 2019 Global Burden of Disease Study estimated that 970 million people, or 12.6% of the global population, live with a mental health disorder, most commonly depression and anxiety [7]. The very high global prevalence of mental health issues makes the study of risk indicators, such as mood, of great importance.

Moods have been shown to vary according to gender, with men tending to report higher vigor and lower anger, confusion, depression, fatigue, and tension than women [8,9]. There are many sociopsychological reasons why women would tend to report more negative moods than men, especially related to the disadvantage women face in many aspects of life, including education, family responsibilities, and careers [10,11]. Gender inequity is evident in Lithuania, which ranked 20 of 27 European Union (EU) countries on the 2021 Gender Equality Index [12], providing a compelling rationale for examining gender differences in mood in our study. 

Age has also been implicated in mood differences, wherein reported moods among adults tend to be more positive with increasing age [7,8]. Differential use of emotion-regulation strategies may partially explain age-related variations in mood. For example, compared to older adults, younger adults tend to engage more in maladaptive coping strategies, such as rumination, avoidance, and suppression [13]. Similarly, mindfulness facilitates effective emotion regulation [14] and psychological wellbeing [15], and older adults are more likely to be classified into high mindfulness groups than their younger counterparts [16]. Hence, it is appropriate to explore the influence of age on mood in the present study. 

A process known as mood profiling, in which scores on a mood scale are plotted against normative scores to create a graphical profile, has a long history of use in identifying common patterns in mood states [17,18]. The Brunel Mood Scale (BRUMS), originally developed in England for use with male and female adolescents and adult athletes and nonathletes, is frequently used to assess mood in research and applied settings due to its short administration time (2–3 min.) and acceptable psychometric characteristics [19,20]. For example, during validation studies, the factorial, convergent and divergent validity of the BRUMS and the invariance of its measurement model was demonstrated among four samples of participants (*n* = 2,549), independently and via multi-sample analysis [20]. 

The 24-item BRUMS was developed as a shortened version of the 65-item Profile of Mood States (POMS) [21,22] to be suitable for adolescents, as well as adults. Hence, items for consideration were screened for comprehensibility by adolescents aged 12 and over, prior to inclusion in an initial list of 42 items, which was reduced to 24 items, following tests of factorial validity (see [19,20] for further details of the validation process). Compared to the POMS, the BRUMS has a shorter completion time (2–3 min vs. 7–10 min), is suitable for adolescents, as well as adults, and does not include mood descriptors, such as “blue” and “bushed”, that may not be understood in all cultural contexts [19]. 

The POMS was originally developed for use with psychiatric outpatients [21] and is, hence, oriented towards negative rather than positive moods, with six subscales representing the mood clusters of Anger-Hostility, Confusion-Bewilderment, Depression-Dejection, Fatigue-Inertia, Tension-Anxiety and Vigor-Activity [21]. The POMS has been criticized for its negative orientation and for providing a limited assessment of mood rather than a comprehensive measure of the mood construct [23], although it has been shown to have utility in screening for mental health issues [24] and predicting outcomes in competitive domains, especially sport, where negative moods can debilitate performance [18]. In terms of screening for mental health issues, negative mood is a risk factor for several psychopathologies [6]. Morgan’s mental health model [24], which is based on the POMS, posits that a mood profile of high Vigor combined with low Anger, Confusion, Depression, Fatigue, and Tension is indicative of positive mental health, whereas a mood profile of low Vigor combined with high Anger, Confusion, Depression, Fatigue, and Tension is associated with poor mental health outcomes. 

In a mental health context, the BRUMS has been used to help manage performance anxiety and prevent injuries among adolescent ballet dancers in Japan [25]; screen for risk of post-traumatic stress disorder among military personnel in South Africa [26]; assess adolescents for elevated suicide risk in the USA [27]; and evaluate population-level mental health and monitor the psychological wellbeing of cardiac rehabilitation patients in Brazil [28,29]. Related to use of mood profiling for predicting performance, Lane and Terry [1] presented a conceptual model that emphasized the interactive effects of different mood dimensions on performance, particularly emphasizing how depressed mood interacts with anger and tension. For example, in the absence of any symptoms of depressed mood, anger has been shown to associate with good performance [30,31], whereas anger associates with poor performance when symptoms of depressed mood are evident [18]. Evidence also exists that negative mood can have positive effects on some types of performance, such as increasing cognitive and emotional creativity via self-focused attention [32]. 

Using mood profiling, several distinct patterns of mood scores have been identified. For example, the iceberg profile, which is characterized by a high Vigor score combined with low scores for Tension, Depression, Anger, Fatigue, and Confusion, typically associates with positive mental health and good athletic performance [18,24,33,34]. The iceberg profile was so named because the shape of the profile is reminiscent of an iceberg, in that most of the scores sit below the waterline (i.e., the mean standardized score of 50), with only the vigor score above the surface, in the same way that most of a real iceberg sits below the surface. The inverse iceberg profile, which is characterized by a below average Vigor score, combined with above average Tension, Depression, Anger, Fatigue, and Confusion scores, typically associates with increased risk of pathogenesis and underperformance [35,36]. Similarly, the inverse Everest profile, the most negative profile characterized by a low Vigor score, high scores for Tension and Fatigue, and very high scores for Depression, Anger, and Confusion indicates elevated risk of disorders, including post-traumatic stress disorder [26]. 

The shark fin profile, which is characterized by below-average scores for Tension, Depression, Anger, Vigor, and Confusion, combined with very high Fatigue scores, may signal “an accident waiting to happen” and has been linked to athletic injury [37] and poor adherence to safety procedures in high-risk vocations [38]. The surface profile, which is characterized by average scores on all mood dimensions, can be considered to represent an average mood, and the submerged profile, which is characterized by below-average scores on all mood dimensions, may be beneficial in activities that place a premium on remaining calm and unemotional [39]. These six mood profiles have been replicated in a variety of language and cultural contexts, including English [8,40,41], Brazilian [42], Chinese [43], Italian [44], and Singaporean populations [45,46]. 

Several validated translations of the BRUMS exist, including Afrikaans [47], Brazilian Portuguese [48], Chinese [49], Czech [50], French [51], Hungarian [52], Italian [52,53], Japanese [54], Malay [55,56], Persian [57], Serbian [58], and Spanish [59]. In most instances, the BRUMS measurement model was supported in translation [47,48,50,51,52,53,54,55,56,58], although in a few translations, some poor-fitting items were discarded [49,57,59]. To date, no validated translation of the BRUMS into the Lithuanian language has been published. Preservation of the Lithuanian language and translation of English-language resources into Lithuanian are seen as significant imperatives for the country [60]. Further, mental health issues are of particular concern in Lithuania. For example, subjective wellbeing is lower in Lithuanian society than the EU average [61] and mental health resources, policies, and practices are perceived as being in urgent need of improvement in the country [62,63]. The availability of simple, quick, and effective risk indicators of mental health status, such as the BRUMS translated into the Lithuanian language, represents an important forward step. 

The purpose of our study was to validate a Lithuanian-language version of the BRUMS to facilitate further mood research in a Lithuanian context. It was hypothesized that the 24-item, 6-factor measurement model of the BRUMS (Tension, Depression, Anger, Vigor, Fatigue, Confusion) would be supported in the translated version, as it has been in previous translations [47,48,50,51,52,53,54,55,56,58]. As well as evaluating the hypothesized measurement model of the translated scale (i.e., factorial validity), we also tested its convergent validity (i.e., whether subscale scores correlated highly with concurrent measures of similar constructs) and divergent validity (i.e., whether subscale scores showed minimal correlation with concurrent measures of dissimilar constructs) [64]. More specifically, as reported previously, it was hypothesized that Tension, Depression, Anger, Fatigue, and Confusion scores would correlate highly with measures of Perceived Stress [37,65] and Neuroticism [66,67], the latter of which is described as the propensity to experience negative mood states [67], and would correlate inversely, but to a lesser degree, with Agreeableness, which is described, in part, as being in control of anger and other negative emotions [67]. It was also hypothesized that Vigor, characterized by being energetic and alert, would correlate, at least moderately, with Extraversion, characterized by talkativeness, sociability, and outgoingness, and show strong inverse correlations with Perceived Stress and Neuroticism [66,67]. Finally, it was hypothesized that mood scores would tend to show minimal correlation with measures of Conscientiousness, characterized by self-control, industriousness, responsibility, and reliability [68], and with Openness, characterized by being imaginative, curious, and open-minded [68]. Given that reported mood and the prevalence of mood disorders vary according to gender [7,8], we tested the invariance of the BRUMS measurement model of the Lithuanian translation across genders and compared the mood scores of men and women. We also assessed relationships between mood scores and age.

## 2. Materials and Methods

### 2.1. Participants

In total, 746 individuals participated in the study, comprising 199 who identified as cisgender men (26.7%) and 547 who identified as cisgender women (73.3%). Participants ranged in age from 17 to 79 years (*M* = 41.8 ± 11.4 years). Most participants were tertiary educated, with 583 (78.2%) reporting being a university graduate, or else having some college (*n* = 60, 8.0%) or professional (*n* = 37, 5.0%) training and only 60 (8.0%) reporting a high school diploma and 6 (0.8%) not completing high school. Most participants (*n* = 457, 61.3%) reported residing in Lithuanian cities larger than 100,000 residents with the remainder living in cities of less than 100,000 residents (*n* = 172, 23.1%), towns of 500 to 3000 residents (*n* = 66, 8.8%), or villages (*n* = 51, 6.8%). Concerning employment, most participants reported sedentary-type vocations (*n* = 570, 76.4%). The remaining participants reported vocations with lots of movement (*n* = 163, 21.8%) and a small number in hard physical labor vocations (*n* = 13, 1.7%). Finally, in terms of self-reported health status, participants reported being in great (*n* = 133, 17.8%), good (*n* = 420, 56.3%), satisfactory (*n* = 173, 23.2%), or poor (*n* = 20, 2.7%) health relative to others of their age.

### 2.2. Measures

#### 2.2.1. Brunel Mood Scale

The Brunel Mood Scale (BRUMS) is a 24-item measure of mood developed originally for use with athletes and adolescents, and since validated for use with a wide variety of populations [19,20]. Originally adapted from the Profile of Mood States (POMS) [21,22], the measure has six subscales of four items each (i.e., Tension—items nervous, anxious, worried, panicky; Depression—items unhappy, miserable, depressed, downhearted; Anger—items bitter, angry, annoyed, energetic; Vigor—items energetic, active, lively, alert; Fatigue—items exhausted, tired, worn out, sleepy; and Confusion—items mixed up, muddled, uncertain, confused). Participants respond on a 5-point Likert scale of 0 = not at all, 1 = a little, 2 = moderately, 3 = quite a bit, and 4 = extremely, with total possible subscale scores ranging from 0–16. The standard response timeframe is to ask respondents “How do you feel right now?” All subscales have shown satisfactory internal consistency, with Cronbach alpha coefficients ranging from 0.74 to 0.90 [19,20]. In the original validation studies, the BRUMS demonstrated adequate psychometric properties using multi-sample confirmatory factor analysis that supported the configural, metric, scalar, and residual invariance of the measurement model across samples of adult students, adult athletes, young athletes, and schoolchildren [19,20]. 

As well as producing subscale scores, an overall mood score, referred to as a Total Mood Disturbance (TMD) score, can be calculated by summing the scores for Tension, Depression, Anger, Fatigue and Confusion and then subtracting the Vigor score [21,22]. However, the developers of the BRUMS [19,20] do not recommend calculating this score, because (a) combining six scores into one represents an unnecessary loss of information, and (b) the TMD score treats Tension and Anger as inherently negative mood states, whereas studies have shown them to be facilitative of some types of performance, such as in combat sports [30,31]. 

#### 2.2.2. Perceived Stress Scale

The Perceived Stress Scale (PSS) is a 10-item measure of personal stress levels [69]. Respondents indicate how often during the past month they have, for example, “felt they were unable to control the important things in life” on a 5-point Likert scale from 0 = never, 1 = almost never, 2 = sometimes, 3 = fairly often, and 4 = very often. Four of the 10 items are reverse scored. The range of possible scores is 0–40, with higher scores indicating higher perceived stress. Scores ranging from 0–13 indicate low stress; scores ranging from 14–26 indicate moderate stress, and scores ranging from 27–40 indicate high perceived stress [69]. Internal consistency coefficients for the PSS among groups tested in the original validation studies of the scale ranged from 0.83 to 0.90 [69]. The Lithuanian-language version of the PSS [70] was used in the present study to assess the concurrent validity of the BRUMS-LTU and showed satisfactory internal consistency (Cronbach alpha = 0.86). The PSS was chosen as a concurrent measure in the present study because perceived stress has been shown to correlate positively with mood scores for Anger, Confusion, Depression, Fatigue, and Tension, and inversely correlate with Vigor scores [37], facilitating a test of convergent validity. 

#### 2.2.3. Big Five Personality Test

The Big Five Personality Test (Big Five) [68,71] is a commonly used measure of personality [72,73]. The measure has 44 items to assess Extraversion, encompassing traits such as being talkative, energetic, and assertive (8 questions); Neuroticism, encompassing traits such as being tense, moody, and anxious (8 questions); Conscientiousness, encompassing traits such as being organized, thorough, and planful (9 questions); Agreeableness, encompassing traits such as being sympathetic, kind, and affectionate (9 questions), and Openness to Experience, encompassing traits such as being imaginative, insightful, and having a wide range of interests (10 questions). Respondents rate their agreement to statements about themselves on a Likert scale where *1 = disagree, 4 = neutral, and 7 = agree*. The internal consistency of the five subscales was supported in the original validation studies, with alpha coefficients ranging from 0.76 to 0.88 [68]. In the present study, we used the Lithuanian-language version of the Big Five [74] to assess concurrent validity of the BRUMS-LTU. Cronbach alphas were acceptable (Neuroticism = 0.88, Extraversion = 0.73, Agreeableness = 0.72, Conscientiousness = 0.78, Openness = 0.64). The Big Five test was chosen as a concurrent measure in the present study because the Neuroticism scale has been shown to correlate strongly with negative mood scores [66,67], the Extraversion scale to correlate moderately with positive mood, in this case, Vigor scores [67], whereas Conscientiousness and Openness scores tend to show minimal correlation with negative mood scores [68], allowing both convergent and divergent validity to be tested. 

### 2.3. Procedure

A native Lithuanian research team had previously developed the BRUMS-LTU by following a translation-back translation process of the BRUMS for use in experimental brain research [75,76], although no psychometric properties for the translated scale were previously reported. This research team provided their translated questionnaire, which is shown in Appendix A, for the current validation study. The BRUMS-LTU was distributed to potential participants as a Google Form online questionnaire via Facebook with the intention of recruiting a large adult Lithuanian sample. Data collection occurred over a 4-month period from 29 May to 28 September 2021. Participation was anonymous and no personal information (e.g., name, email addresses) was collected during the questionnaire process. Participants were presented with the research purpose and ethical approval (No. STIMC-BTMEK-08), and by clicking “continue”, provided their informed consent. The protocol was approved by the Human Research Ethics Committee at the Vytautas Magnus University in accordance with the Declaration of Helsinki.

### 2.4. Data Analysis

Analyses were conducted using IBM SPSS and AMOS Statistics for Windows, Version 27 [77,78]. The factorial validity of the BRUMS-LTU was tested using confirmatory factor analysis (CFA), a theory-driven method that utilizes a range of estimation techniques and measurement fit indices to test how well the hypothesized measurement model fitted the sample covariance matrix. Good-fitting models tend to produce consistent results on different measures of model fit [79]. The maximum likelihood (ML) method was used, which specified that items were related to their hypothesized factor with the variance of the latent factor fixed at 1.0. ML is less sensitive to distribution misspecification [80] and performs well over other normal-theory-based methods for large samples [79]. In line with the validation process used previously, the latent factors of Anger, Confusion, Depression, Fatigue, and Tension, were allowed to inter-correlate positively, and to correlate negatively with Vigor [19,20]. 

Several fit indices were used to evaluate the adequacy of the measurement model. Initially, the χ2 to degrees of freedom ratio was considered (where a ratio of <3 represents acceptable fit [81]). However, the χ2 value is always significant for samples with ≥400 cases, and hence was inappropriate as the primary statistic for our purpose [81]. Instead, we gave priority to two incremental fit indices, the non-normed or Tucker–Lewis index (TLI) [82] and the comparative fit index (CFI) [83], which both adjust for the issues of sample size inherent in the χ2 test of model fit. In both instances, values ≥ 0.90 indicate an acceptable fit and values ≥ 0.95 indicate a good fit. We also used the root mean square error of approximation (RMSEA) [84], which indicates the mean discrepancy between the observed covariances and those implied by the model per degree of freedom, thus also avoiding issues related to larger samples. RMSEA values ≤ 0.05 indicate a good fit and values ≤ 0.08 indicate an acceptable fit [85]. Finally, we used the root mean square residual (SRMR), a measure of the average of the standardized fitted residuals, where a value of ≤ 0.08 is indicative of an acceptable model [79]. Our sample of 746 participants exceeded the recommended minimum sample size of 10 participants per model parameter for confirmatory factor analysis [81].

During the data collection period, the number of new COVID-19 cases in Lithuania fluctuated considerably from month to month. June 2021 saw a steady decline in case numbers, which remained low during most of July 2021, before increasing steadily during August 2021, and then increasing more rapidly during September 2021. These fluctuations in case numbers, combined with the associated restrictions on travel and other important aspects of life, had the potential to influence mood scores. Therefore, to test this possibility, a one-way MANOVA was run to compare mood scores by month of data collection.

## 3. Results

### 3.1. Descriptive Statistics

Descriptive statistics for the BRUMS-LTU subscales are shown in Table 1. The full range of possible scores (range = 0–16) was reported for all subscales, with the single exception of Tension (range = 0–15), where no participant reported the maximum score. Some non-normality was evident in the dataset. Positive skewness was apparent for the Anger, Confusion, Depression, and Tension scores, indicating a high proportion of low scores and a long tail towards the upper end of these scales, which is frequently found for measures of negative moods [19,20]. Kurtosis values were high for Anger, Confusion, and Depression scores, which again, is commonly found for theses subscales [19,20]. The Mahalanobis statistic identified 15 significant multivariate outliers (*p* < 0.001) among the data. However, a case-by-case inspection found no examples of response bias in the form of acquiescent, extreme, or straight-line responding [86,87] and, therefore, no data transformations occurred. Reliability coefficients (Cronbach alpha) for the six subscales of the BRUMS-LTU varied from 0.83 to 0.89, exceeding the benchmark for acceptability [79]. 

### 3.2. Confirmatory Factor Analysis

Results of our evaluation on the adequacy of the BRUMS-LTU measurement model are shown in Table 2 and Table 3. The fit of the default six-factor model to the observed covariances fell just short of adequacy benchmarks [82,83,84,85]. Modification indices showed that the measurement model could be improved significantly by modelling covariance between the error terms for two Anger items (annoyed and angry), two Confusion terms (confused and mixed up), two Depression terms (depressed and downhearted) and two Tension items (worried and anxious). All these covariance pathways have been identified in previous validation studies [19,20] and Byrne supported the use of covariance paths between error terms to account for potential overlapping content [88]. The modified six-factor measurement model showed adequate fit on all indices (Table 2). All items loaded on target factors, with 20 of the 24 loadings (83.3%) above 0.70, supporting the measurement model (Table 3). Confirmatory factor analysis was also conducted on gender subsamples to test the measurement model independently in men and women. Fit indices were adequate for both men and women (Table 2). 

Multi-sample analysis was conducted to test measurement invariance by gender. Given the sensitivity to sample size of the change in χ2 statistic, the change (Δ) in other fit indices was used to evaluate invariance at each step [89,90]. Values of ΔCFI ≤ −0.01, ΔRMSEA < +0.015, and ΔSRMR < +0.03 from the unconstrained model to progressively more constrained models supported multiple-group measurement invariance [91]. Furthermore, model fit differences were judged to be nonsignificant if they had overlapping 90% RMSEA confidence intervals [92]. Using these fit criteria, configural, metric, scalar, and residual invariance of the BRUMS-LTU measurement model was supported (Table 4).

### 3.3. Convergent and Divergent Validity

Concurrent measures (i.e., PSS, Big Five) facilitated assessment of both convergent and divergent validity. Inter-correlations among measures for men and women are shown in Table 5. Given our relatively large sample, most correlation coefficients were statistically significant, although many of the observed relationships were small effects (*r* ≤ 0.20) [79]. As hypothesized, the negatively oriented subscales of the BRUMS-LTU (Tension, Depression, Anger, Fatigue, and Confusion) showed strong positive correlations (≥0.50) [79] with Perceived Stress and Neuroticism, for both men and women. Conversely, the Vigor subscale showed moderate-to-strong inverse relationships with Perceived Stress and Neuroticism, and a positive association with Extraversion. These relationships have previously been reported in the literature [37,65,66,67] and provide evidence of convergent validity of the BRUMS-LTU. As hypothesized, negative mood scores showed small correlations (*r* ≤ 0.20) [79] with Conscientiousness and Openness scores in 14 of 20 bivariate relationships (70%) and none of the remaining relationships explained more than 7.8% of variance. Similar relationships have previously been reported [68] and provide evidence of divergent validity in the BRUMS-LTU.

### 3.4. Mood Scores by Gender

A between-group MANOVA was conducted to determine if mood scores varied by gender. Partial *η*² was used as the index of effect size, for which small, medium, and large effects are designated as 0.0098, 0.0588 and 0.1379, respectively [79]. A significant multivariate effect was found for gender, in the small-to-moderate range (Table 6). Follow-up univariate tests showed that Fatigue scores were significantly higher among women, whereas Vigor scores were higher among men. Women reported higher scores than men for Tension, Depression, Anger, and Confusion, although the differences were not statically significant.

### 3.5. Mood Scores by Age

A Pearson correlation matrix was calculated to assess relationships among BRUMS-LTU and age (Table 5). Among men, all correlations between mood scores and age were small effects (r ≤ 0.20) [79]. Among women, four of six correlations between mood scores and age (66.7%) were small effects (r ≤ 0.20) [79] and neither of the other two relationships (tension and age: r = 0.21; fatigue and age: r = 0.24) explained more than 5.8% of variance. Overall, correlations indicated a weak linear relationship between mood and age in the current sample.

## 4. Discussion

The central purpose of our research was to validate the Lithuanian translation of the Brunel Mood Scale questionnaire [19,20], which we refer to as the BRUMS-LTU, among a sample of 746 participants who were fluent in Lithuanian. Results of our analyses supported the factorial validity of the measurement model and its configural, metric, scalar, and residual invariance across gender groups. It appears not only that the factor structure of the BRUMS-LTU is the same in men and women, but also that men and women understand the scale and the construct of mood in a similar way, and that the scale has similar error across genders. The convergent and divergent validity of the scale was adequately supported, as was the internal consistency of the six subscales. Overall, the psychometric properties of the BRUMS-LTU are very closely aligned to those of the original English-language BRUMS [20]. Support for the psychometric integrity of the measure offers considerable potential for using the BRUMS-LTU in future research studies set in a Lithuanian context, including testing Morgan’s mental health model [24] and Lane and Terry’s conceptual model of mood and performance [1]. 

Another future research direction would be to interrogate Lithuanian data using cluster analysis for evidence of the six distinct mood profiles that have been identified previously in other language and cultural contexts; namely, the iceberg, inverse Everest, inverse iceberg, shark fin, submerged, and surface profiles [38,40,41,42,43,44,45]. Given that the full range of possible scores (range = 0–16) was reported for all BRUMS-LTU subscales, with the single exception of Tension (range = 0–15), it is apparent that some participants reported extremely negative moods, which would indicate increased risk of mental health issues. More specifically, it would be anticipated that anyone reporting an inverse Everest profile, characterized by a low Vigor score, high scores for Tension and Fatigue, and very high scores for Depression, Anger, and Confusion, would be a candidate for follow-up assessment by a clinical psychologist or other health professional. Exploring use of the BRUMS-LTU as a mental health screening tool may be a fruitful avenue for future research in Lithuania, considering that the BRUMS has frequently been used for this purpose in English-speaking countries [26,27,28,37].

The benefits of regular physical exercise on psychological wellbeing, including the maintenance of positive mood and prevention of negative mood, is well documented [93,94] and has been shown to be especially pertinent in several countries during the COVID-19 pandemic [95,96]. Given that psychological wellbeing is below the EU average among the Lithuanian population [62,63], there is much scope for intervention studies that evaluate the impact of physical activity programs on wellbeing indicators, including mood, of participants. Further, several applications of mood profiling in the sport domain have been proposed, including to monitor emotional recovery from injury and responses to training load, to identify overtrained athletes, to predict athletic performance, and as a catalyst for discussion with a psychology professional (see [97] for a review and [98] for other applications). Prior to the current validation of the BRUMS-LTU, there was no validated mood scale available in the Lithuanian language, offering fertile ground for a wide variety of research questions and potential applications to be addressed in the sport and exercise domains. 

Gender comparisons identified significant differences in mood scores in the present Lithuanian sample that are largely consistent with previous findings. The higher scores for Vigor and lower scores for Fatigue reported by men compared to women are consistent with the findings of several previous studies, although often, gender differences in mood scores extend to other mood dimensions, particularly Tension, Depression, and Confusion [8,9,44,45]. The relationship between mood scores and the age of participants was in the small effect range and, therefore, inconsistent with previous findings [7,8]. There is no obvious explanation for this lack of association and future research might investigate this relationship further in a Lithuanian context.

Although we provided preliminary evidence of acceptable psychometric characteristics of the BRUMS-LTU, further exploration of its psychometric integrity is recommended. The fit of the default measurement model was shown to be only just acceptable and was improved by adding covariance paths to the error terms for some items, as recommended by Byrne [88], as a legitimate strategy to account for potential overlap in meaning between mood descriptors (e.g., anxious and worried). Further exploration of the measurement model is warranted, and additional psychometric properties of the BRUMS-LTU should be explored, including its test–retest reliability and predictive validity. Several other future research directions can also be identified. For example, to explore the lived experiences of Lithuanians during the COVID-19 pandemic, researchers could use the BRUMS-LTU together with retrospective pandemic experience questions (e.g., self-perceptions of how moods were affected by extended lockdowns or by personal experience of COVID-19, perhaps through contracting the disease personally or vicariously through relatives or close friends having caught the disease). Extending the validation of the BRUMS-LTU to youth participants would facilitate additional opportunities for impactful mood research, across many contexts, such as education and sport. Finally, once sufficient additional data are available from Lithuanian participants, tables of normative scores for the BRUMS-LTU should be generated to provide a reference point for interpreting data derived from the scale. Given the gender difference identified in the present study, separate norms for men and women may be required.

In common with all online questionnaire research, limitations were evident in our study. One limitation relates to our lack of knowledge of whether participants’ lived experiences during the COVID-19 pandemic influenced their moods, as has been shown in previous research [9]. The observed lack of significant differences in mood scores during different phases of the pandemic suggests that mood did not fluctuate greatly during our data collection period. A second limitation concerns the requirement to access the questionnaire via the internet. Online questionnaires of any kind reduce access by marginalized and lower socio-economic groups. Thus, our BRUMS-LTU was validated on a predominantly university-educated sample, although it should be noted that 54% of the Lithuanian population hold a tertiary qualification, which is among the top 10 graduate rates in the world [99]. A third limitation relates to the age of participants. Although the mean age of our sample (41.8 years) approximated the 2020 median age of 44.5 years for Lithuania as a whole [100], we did not assess the moods of anyone under 17 years of age. A fourth limitation relates to the gender representation of our sample, which included 73.3% of participants who identified as women. Finally, although the translation process followed the recommended translation-back translation process [101], this is seen as a minimum standard in good practice guidelines for test adaptation [102], and the semantic and measurement equivalence of the translated BRUMS to the original English-language version needs further examination. Collectively, these limitations may restrict the ability to generalize our findings to the broader Lithuanian population. 

## 5. Conclusions

In conclusion, it is apparent that the BRUMS-LTU demonstrated satisfactory psychometric characteristics and that gender was associated with differences in mood among Lithuanian respondents. The BRUMS-LTU provides an appropriate measure with which to further investigate moods among Lithuanian participants.

## Figures and Tables

**Table 1 ijerph-19-04867-t001:** Descriptive statistics for BRUMS-LTU subscales (*n* = 746).

Variable	*M*	*SD*	*SE_M_*	Min	Max	Skewness	Kurtosis	α
Anger	2.35	3.20	0.12	0	16	1.70	2.70	0.86
Confusion	2.84	3.46	0.13	0	16	1.41	1.55	0.85
Depression	2.62	3.53	0.13	0	16	1.54	1.75	0.88
Fatigue	5.12	4.22	0.15	0	16	0.59	−0.59	0.89
Tension	3.43	3.56	0.13	0	15	1.10	0.51	0.83
Vigor	9.13	3.70	0.14	0	16	−0.20	−0.50	0.88

**Table 2 ijerph-19-04867-t002:** Model testing of the BRUMS-LTU.

Group	χ2	*df*	χ2: *df*	CFI	TLI	RMSEA	90% CI	SRMR
Six-factor default model (*n* = 746)	1188.20 *	235	5.06	0.928	0.916	0.074	[0.070, 0.078]	0.088
Six-factor modified model (*n* = 746)	833.44 *	226	3.69	0.954	0.944	0.060	[0.056, 0.064]	0.070
Six-factor modified model-men (*n* = 199)	500.58 *	226	2.22	0.927	0.911	0.078	[0.069, 0.088]	0.073
Six-factor modified model-women (*n* = 547)	694.51 *	226	3.07	0.952	0.941	0.062	[0.056, 0.067]	0.068

Note. CFI = Comparative fit index, TLI = Tucker–Lewis index, RMSEA = Root mean square error of approximation, CI = Confidence interval, RMSR = Standardized root mean square residual. * *p* < 0.001. The six-factor modified model allowed covariance between the error terms for two Anger items (annoyed and angry), two Confusion terms (confused and mixed up), two Depression terms (depressed and downhearted) and two Tension items (worried and anxious).

**Table 3 ijerph-19-04867-t003:** Standardized factor loadings for the BRUMS-LTU (*n* = 746).

Item	Tension	Depression	Factor Anger	Vigor	Fatigue	Confusion
Nervous	0.796					
Anxious	0.786					
Worried	0.742					
Panicky	0.549					
Unhappy		0.850				
Depressed		0.842				
Downhearted		0.838				
Miserable		0.694				
Bad tempered			0.868			
Bitter			0.810			
Angry			0.738			
Annoyed			0.633			
Energetic				0.889		
Active				0.832		
Lively				0.805		
Alert				0.709		
Exhausted					0.897	
Tired					0.867	
Worn out					0.844	
Sleepy					0.675	
Mixed up						0.803
Muddled						0.763
Uncertain						0.744
Confused						0.724

**Table 4 ijerph-19-04867-t004:** Fit indices of invariance models for men vs. women.

Invariance Model	χ2	*df*	χ2: df	CFI	ΔCFI	TLI	RMSEA	ΔRMSEA	90% CI	SRMR	ΔSRMR
Configural	1195.67 *	452	2.65	0.945		0.933	0.047		[0.044, 0.050]	0.065	
Metric	1235.28 *	476	2.60	0.944	−0.001	0.935	0.046	−0.001	[0.043, 0.049]	0.066	+0.001
Scalar	1291.53 *	500	2.58	0.941	−0.003	0.935	0.046	0.000	[0.043, 0.049]	0.068	+0.002
Residual	1432.08 *	550	2.60	0.934	−0.007	0.934	0.046	0.000	[0.043, 0.049]	0.069	+0.001

Note. Δ = change in, CFI = Comparative fit index, TLI = Tucker–Lewis index, RMSEA = Root mean square error of approximation, CI = Confidence interval, RMSR = Standardized root mean square residual. * *p* < 0.001.

**Table 5 ijerph-19-04867-t005:** Inter-correlation matrix of BRUMS-LTU, PSS, Big Five subscales and age. Men (*n* = 199) upper right of matrix. Women (*n* = 547) lower left of matrix.

Variable	1	2	3	4	5	6	7	8	9	10	11	12	13
1. Tension		0.82	0.73	−0.40	0.60	0.80	0.56	−0.32	−0.40	−0.28	0.64	−0.20	−0.10
2. Depression	0.80		0.77	−0.43	0.61	0.85	0.61	−0.37	−0.37	−0.24	0.64	−0.18	−0.09
3. Anger	0.77	0.76		−0.39	0.61	0.72	0.59	−0.26	−0.43	−0.23	0.64	−0.18	0.04
4. Vigor	−0.34	−0.42	−0.32		−0.58	−0.36	−0.52	0.42	0.38	0.39	−0.60	0.34	−0.01
5. Fatigue	0.62	0.62	0.55	−0.58		0.54	0.50	−0.32	−0.27	−0.15	0.50	−0.17	−0.16
6. Confusion	0.76	0.74	0.69	−0.33	0.54		0.59	−0.33	−0.39	−0.25	0.58	−0.22	−0.11
7. Perceived Stress	0.63	0.66	0.57	−0.44	0.53	0.55		−0.35	−0.48	−0.43	0.76	−0.33	−0.07
8. Extraversion	−0.27	−0.34	−0.29	0.44	−0.32	−0.31	−0.34		0.25	0.35	−0.38	0.50	0.03
9. Agreeableness	−0.25	−0.21	−0.27	0.24	−0.18	−0.20	−0.32	0.17		0.50	−0.63	0.34	0.15
10. Conscientiousness	−0.11	−0.15	−0.12	0.30	−0.17	−0.19	−0.24	0.28	0.41		−0.43	0.34	−0.05
11. Neuroticism	0.60	0.59	0.52	−0.48	0.48	0.53	0.76	−0.33	−0.46	−0.29		−0.30	−0.08
12. Openness	−0.13	−0.18	−0.22	0.30	−0.17	−0.10	−0.29	0.42	0.32	0.28	−0.31		0.07
13. Age	−0.21	−0.20	−0.14	0.18	−0.24	−0.19	−0.30	0.14	0.21	0.11	−0.26	0.29	

Note. Men: correlations > 0.25 are *p* < 0.001, >0.15 are *p* < 0.05, others are non-significant; Women: correlations > 0.13 are *p* < 0.001, others are *p* < 0.01.

**Table 6 ijerph-19-04867-t006:** Comparison of BRUMS-LTU raw scores by gender (*n* = 746).

Source	*n* (%)	*Mean*	*SD*	*F*	Sig. Diff.	α
Gender [Wilks’Λ = 0.973, *F*(6, 739) = 3.46 **, partial *η*² = 0.027]						
Men (M)	199 (26.7)					
Tension		3.14	3.46	1.77	N.S.	0.82
Depression		2.37	3.33	1.36	N.S.	0.91
Anger		2.28	3.13	0.12	N.S.	0.87
Vigor		9.95	3.43	13.64 ^†^	M > W	0.87
Fatigue		4.24	3.76	12.03 ^†^	W > M	0.88
Confusion		2.55	3.24	1.90	N.S.	0.85
Women (W)	547 (73.3)					
Tension		3.53	3.59			0.83
Depression		2.71	3.60			0.88
Anger		2.37	3.23			0.86
Vigor		8.83	3.76			0.88
Fatigue		5.44	4.34			0.89
Confusion		2.95	3.53			0.85

Note: ** *p* = 0.002, ^†^
*p* < 0.001. N.S. = not significant, *p* > 0.05.

## Data Availability

Data are available from the corresponding author.

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
