# Peer review of "Validation of a Lithuanian-Language Version of the Brunel Mood Scale: The BRUMS-LTU"

_ijerph, 2022, doi:10.3390/ijerph19084867_

Round 1
Reviewer 1 Report
The manuscript is well written. I hope the following suggestions are helpful for the authors to further enhance the quality of the manuscript.
Introduction
- p. 1, the authors referred the Lithuanian-language version of the Brunel Mood Scale (BRUMS) to the Lithuanian Mood Scale (LTUMS). I wonder when the name occurs in the future literature will create an impression that it is a new measurement rather than the translated version of the BRUMS.
- p. 1, citations are needed to support the following statement:
"very negative moods often signal increased risk of mental health disorders, including depression, anxiety, bipolar disorder, eating disorders, schizophrenia, and drug or alcohol abuse"
Materials and Methods
- p. 3, elaborate on the robust psychometric properties that have been found in the original validation studies.
- The introduction of the several previous translations and validations of the BRUMS is better to be presented in the Introduction part. Moreover, it is essential to indicate whether those studies found consistent (or inconsistent) findings.
- p. 3, if the authors are reporting the different language versions of the BRUMS have been developed and validated, take note that Singaporean is not a language. The authors shall refer to the article and verify whether the English version of the BRUMS was validated in a sample of Singaporeans.
- Explanations are needed to justify the use of the Perceived Stress Scale and Big Five Inventory scores in testing concurrent validity.
- p. 4, "the χ2 to degrees of freedom ratio was considered (where a ratio of < 3 represents acceptable fit; [44], although this index always produces a statistically significant value". The writing is inaccurate because a p-value is generated for the chi-square test result but not the (χ2 to degrees of freedom) ratio.
- What is the reason to include the "The Lithuanian Mood Scale (LTUMS)" section? Is it a part of Figure 1?
- It is intriguing to know whether any models other than the six-factor model have been found in the literature. If yes, kindly examine the competing models. In addition, with the sample size, the authors may consider splitting the data into two halves to conduct exploratory and confirmatory factor analysis respectively.
Results
- Please check and report if the multivariate normality is supported.
- Table 2. As the CFI value suggests the model is good fit while the other two indicators suggest the model has an acceptable fit, the authors are suggested to report the SRMR value (or other values) to offer a more comprehensive evaluation of the model fit.
- Kindly explain the Six-factor modified model in the note of Table 2.
- p. 7, explain the meaning of the z score in the standard formula: T = 50 + (10 * z).
- Explain the scores of the six factors in the note of Table 3.
- Why did not the authors examine the measurement invariance among age groups?
Discussion
- p. 10, it is suggested to present the six mood profiles in the Introduction to introduce the functions of the BRUMS and highlight the need for the development of the LTUMS.
- Suggest reminding readers that the sample size of the female group is much greater than the male group and may influence the comparison result.
- Suggest directions to further examine the psychometric qualities of the LTUMS, especially those (e.g., test-retest reliability, predictive validity) that were not examined in this study.
Reviewer 2 Report
Thank you for giving me the opportunity to read your manuscript “Development and Validation of the Lithuanian Mood Scale (LTUMS)”
I have some concern and suggestions how the manuscript could be improved. My main concerns are summarized and followed by detailed suggestions:
I) Invariance tests across gender are missing.
II) Convergent and divergent validity need to be described clearer (including arguments in the Introduction).
III) Comparing the Lithuanian scores to a English speaking population norm (and transforming scores into T-scores based on the English speaking population norms) does not seem correct. I am not familiar with the Brunel Mood Scale. If T-scores are not essential, I suggest deleting those parts in the manuscript that mention T-scores; if a norm population and the transformation to T-scores are essential for the Mood Scale I do not think the manuscript can be corrected by further revisions, because the data for such an endeavor have not been assessed.
IV) Discussion of gender needs to be included already in the Introduction. Be careful to understand gender in the social-cultural context (and do not “reduce” gender to solely biological concepts).
My detailed concerns and suggestions:
Major Points
Title
1) In the Abstract it is stated that "The current study involved the translation and validation of the Brunel Mood Scale (BRUMS)". Thus, no new Scale was developed. The Title includes the term “development” and is therefore hinting that a new Scale was developed, which does not seem to be the case.
Abstract
2) I suggest giving a more specific definition or understand what moods are (maybe 10.1177/1754073909103594). I think that the first sentence of the Abstract is not informative enough.
3) I would state that measures of mood exist in various languages in the second sentence. The term or expression “mood response” seems confusing (because, as you state in the Introduction, mood is “less clearly related to a specific cause”).
4) What do you mean by “The resultant 24-item” – does the BRUMS have another item structure? Also report results of the study after describing the methods.
5) What is a “first” language. Better use terms such as “were fluent in”, “proficient in”, “a certain level of”.
6) According to the APA manual, but also AMA manual, it is suggested to use “women” “men” when referring to people; and only use “female” “male” as adjectives (such as in “female participants”). Therefore I suggest, “nwomen = 547, nmen = 199” instead of “male = 199, female = 547”; another suggestion would be to give percentages – those would be easier to understand.
7) I would call the statistical method, confirmatory factor analysis (CFA) if you tested a predefined model; I suppose you tested the original BRUMS factor structure.
8) Include the confidence interval for RMSEA.
9) I would refrain from stating “exceeded accepted thresholds”, but instead write “were satisfactory”. You can also report the values.
10) Use “gender” instead of “sex” (see comment 6; APA manual 7th ed, chapter 5.5; https://apastyle.apa.org/style-grammar-guidelines/bias-free-language/gender)
11) Did you do invariance testing?
12) You report some indices for convergent validity. Did you also consider divergent validity? (scales that are not supposed to correlate with certain other scales)
13) Maybe write your conclusion of the Abstract with more sentences. E.g., “Findings offer supported for adequate psychometric properties of the LTUMS. Thus, the LTMUS can be recommended for use in further psychological studies on mood in Lithuania.” – and maybe can be used in practice (what you call on page 2 “applied setting”).
Introduction
13) Maybe “very” in “very negative moods often signal increased risk of mental health disorders”, but rather “persistent negative moods”.
14) Use a reference (citation) after stating that persistent negative mood is linked to psychological problems.
15) I suggest using a peer reviewed source for [3]; or you can delete the sentence altogether.
16) “mood responses” – maybe better use “mood states”. As already said (Comment 3) “response” does better fit the description of emotions.
17) I would not state “has been used to assess mood in more than 200 published studies” – first this value is difficult to interpret (is this much? For a Scale developed in 1999 it does not seem so); second what kind of “published” and how did you find out? Either there is review that summarizes research with BRUMS, or you do not use such a statement.
18) You need to cite a study that shows that the BRUMS is short, and has robust psychometric properties (in what samples?).
19) First report in what language and for what sample the Scale was developed. Then, I suggest citing exemplars; however, it would be best to have a review available on the multiple uses of the scale.
20) I do not think the geographical information on Republic of Lithuania (page 2) is necessary. It would be more relevant to know how mood differs on average and how mood has been conceptualized in Lithuania in comparison to the sample/population from which the BRUMS was developed from.
21) Does the source [14] compare suicide rates in Lithuania with other EU countries? Why do you talk about suicide in specific?
22) You can give the explanation of the link between negative mood and mental health problem more space; then try to use models and explanations for this associations. Just picking some mental health problems seems arbitrary and confusing.
23) You can already in the Introduction explain, what scales we can expect; has mood dimensions/facets?
24) Also at page 2 I would state that you translated, rather than developed a Lithuanian-language version of the BRUMS.
25) Try to already explain in the Introduction, what convergent/divergent validity you expect the scales to have.
26) Try to explain what steps are needed to translate the BRUMS (from English?) to Lithuanian. Use guidelines and suggestions such as 10.1016/j.bodyim.2018.08.014.
Methods
27) Try not to imply value-statements in the description of your sample. Such as, “well educated” (vs. the other are not?), “small village” – why not only “village”?
27.1) Argue how and why your sample size was large enough – include some sample size, effect size, power analyses.
28) Write out the full scales name in the first sentence of 2.1.1. It does not seem sufficient to “define” an aberration in a heading – this applies to other scales and headings as well
29) Be specific in what country the BRUMS was developed in; also whether the sample consisted of women or men; and what is known about invariance across gender or other variables. Also report about gender similarities or differences – and include the discussion of gender in the Introduction.
30) Discuss the POMS and the improvements made with the BRUMS (as you shortly do in 2.1.1.) already in detail in the Introduction.
31) Also does this “POMS measurement model for providing a limited assessment of the global domain of mood rather than a comprehensive measure of the mood construct” mean that the BRUMS does not have a “global” mood score? And what do you mean by “comprehensive”
32) At the end of 2.1.1 you mention “LTMUS” for the first time. You need to define this term when using it for the first time.
33) Furthermore, the sentence “using the LTUMS are cautioned against extrapolating findings beyond the six specific mood dimensions” is confusing, because you argue the sentence before, that the BRUMS is “comprehensive”. Be more consistent and maybe exact with your arguments.
34) A sentence needs to be clear without its parenthesis. This is why the sentence does not seem finished: “Respondents indicate how often they have felt and/or thought during the past month” – felt what?
35) Report originally reported internal consistencies for all scales.
36) The information about Big Five Personality Test’s often use does not seem to be reflected in the offered citations.
37) Shortly explain all “big five” concepts.
38) Why do you report the range of scores of the Big Five scales in the methods?
39) It is not clear what Figure 1 represents. If those are the scale items, rather offer them as appendix or supplemental in doc-format. It is not clear why you call a table a figure. Make sure that all text is readable.
40) Describe how participants were recruited.
41) Was data assessment after lockdown? “This period represented the end of the COVID-19 pandemic second wave, a lengthy low in reported cases, and then a sharp return of a third wave” was the “sharp return” still during data assessment. This part does not seem clear to me.
41) Reword “Good-fitting models tend to produce consistent results on different measures of model fit” rather maybe state “the model needs to fit the data, indicated by consistent results on different measures of model fit”
42) Also use the root mean square residual (SRMR)
43) After 2.3. Data Analysis the chapter “The Lithuanian Mood Scale (LTUMS)” is not in English. If this is part of Figure 1, I suggest to follow Comment 39 and offer the scale ad appendix or supplemental in doc-format.
44) I think you could already report in “2.1.1 Brunel Mood Scale (BRUMS)” that you used the Lithuanian Brunel Mood Scale from another study. Also report the reported psychometric properties by those authors.
Results
45) I would rather use “Descriptive statistics” than “Summary statistics”.
46) Do not state “clearly exceeding the benchmark for acceptability” but rather “internal consistencies were acceptable”
47) Do not discuss results already in the Results section (i.e., “Mean scores for the six subscales generally approximated the English-language normative scores”) but rather describe what mood did participants on average report (e.g., were they in a good/moderate/negative mood).
48) The first mentioning of Figure 2 on page 5 is confusing, because Figure 2 shows standardized scores – standardized in relation to a norm population. This is confusing (1) you do not mention in the paragraph that scores are transformed into T-scores, (2) it does not seem appropriate to use the English speaking norm population. You would have needed to assess your own Lithuanian norm population.
49) Is there not the possibility to calculate a “general” factor? Would it not have been theoretically meaningful to calculate a bifactor model?
50) How can the correlation of error term between items that do not load on the same scale be justified?
51) “All items loaded on target factors with loadings above .50” report all loadings, e.g., in a table.
52) “3.3. Normative Scores for the LTUMS” using a different countries norm scores does not seem justified.
53) “The opposite was true for the Vigor subscale”, namely (spell it out)?
54) “Between-group MANOVAs were conducted to determine if mood scores varied by sex” use “gender”
55) Report about gender similarities and differences in the Introduction.
56) Report means of women and men (including alphas); and then mark when scores are different.
57) Do an invariance analysis across gender to see whether the scales measures the same in women and men.
58) How and why did you group people’s age? Why did you not do a correlation analysis?
59) Include age in the Introduction; report how average mood changes across life span.
60) If you keep age in the age groups justify how you chose those groups; then also report mean scores of each group (including alphas) and mark were there are differences.
Discussion
61) Cite the “Brunel Mood Scale” in the first sentence of your Discusison and explain why you refer to the translation as LRUMS.
62) It does not seem clear what you mean by “concurrent” – is this convergent or divergent validity?
63) “reinforced the internal consistency of the six subscales” do you mean “supported”?
64) “generation of a table of normative scores for the LTUMS” I think this is a major limitation of your study, to not have assessed norm values. It does not seem correct to compare “raw scores” with normative scores from another population (and measure).
65) What are “mood profiles”, and why are they important?
66) Compare your psychometric results to other validation studies – what can be improved? Why do error terms need to correlate?
67) As already stated before, discuss gender already in the Introduction.
68) Do not rely solely on biological concepts for explaining gender differences in mood. Being subject to discrimination and gender disparities that disadvantage women to a higher degree might very easily explain negative mood in woman compared to men – this has nothing to do with the “inability to effectively downregulate negative feelings”
69) It does seem inconsistent that mood should not have gotten on average more negative due to the pandemic.
70) Offer some more reflection on the Limitations, e.g., how has the questionnaire been translated? Did you follow all suggestions by 10.1016/j.bodyim.2018.08.014?
71) “via mobile phones, which are considerably more prevalent in Lithuania than the internet” this sentence does not make sense. First, if your study was on Google forms, your study was accessible on the phone too. And it is the internet access on the mobile phones that allows people to access the internet.
72) What are “positive psychometric characteristics” – do you mean satisfactory?
73) What do you mean by “sex and age group moderated mood responses”?
74) Delete “antecedents, correlates, and behavioral consequences” from the last sentence.
75) Discuss the results of convergent and divergent validity in more detail – also what they mean.
Round 2
Reviewer 2 Report
Thank you for letting me read the revised manuscript of “Validation of a Lithuanian-language version of the Brunel Mood Scale: the BRUMS-LTU”
Even though I think the manuscript has considerably improved, I do not think that the most concerning limitations have been addressed. Namely, there is still no invariance analysis and authors transform scores into T-scores based on their sample – which does not seem representative.
I still think the Introduction and the Discussion need further improvement. First the Introduction focuses on mental health problems whereas those problems are not mentioned in the Discussion. The Introduction does not seem to offer enough information to understand all hypothesis – especially convergent and divergent validity.
My remaining list of Major Points:
Abstract
2) I suggest giving a more specific definition or understand what moods are (maybe 10.1177/1754073909103594). I think that the first sentence of the Abstract is not informative enough.
RESPONSE: The first sentence of the abstract has been modified.
>>> Further suggestion: I would be even more specific and exact in the definition. First the first half of the sentence, that moods are an integral part of human existence does not seem to add much to the definition. I would suggest defining mood to be an emotional state (valence and quality) that is not directed towards a specific object and can be experienced as global state, hence mood.
Avoid references in the Abstract.
11) Did you do invariance testing?
RESPONSE: Invariance testing has been added and is reported in Table 2.
>>> Further suggestion: I do not find invariance tests in Table 2. See for example Kline, R. B. (2016). Principles and Practice of Structural Equation Modeling (4th Ed.). The Guilford Press page 396 (https://www.guilford.com/books/Principles-and-Practice-of-Structural-Equation-Modeling/Rex-Kline/9781462523344). You have to test configural, metric, scalar and residual invariance. See for example other validation studies: 10.1037/cou0000225, 10.1016/j.bodyim.2021.12.002, 10.1037/sgd0000554.
Invariance test can inform you, whether factor structure is the same in women and men (configural), whether they have similar factor loadings (thus, understand the scale in a similar way; metric), and whether women and men have the intercept at a similar point (thus, whether the “starting point” is the same). Scalar invariance is important in order to know whether gender differences in score values might indicate differences in the latent concept. Residual invariance can give information on whether the scale has similar error in women and men.
12) You report some indices for convergent validity. Did you also consider divergent validity? (scales that are not supposed to correlate with certain other scales)
RESPONSE: This is now addressed on p. 4
>>> Further suggestion: Be exact what scales are expected to correlate. Avoid the expression “some subscales” (page 4)
Introduction
15) I suggest using a peer reviewed source for [3]; or you can delete the sentence altogether.
RESPONSE: This section has been rewritten and a peer reviewed source has been added.
>>> Further suggestion: Avoid using the word “normal” (end of page 1)
I am not sure whether “depressed mood” is a symptom, rather than a risk factor, of depression (e.g., doi.org/10.1038/nrdp.2016.65).
Why is there such a focus on disorders?
What are “positive moods” – happiness? Explain that moods can have a valence (positive vs. negative) and quality (happy, irritated, angry).
Do not reduce women to their hormonal level! There is much sociopsychological reason why women might face discrimination and disadvantage in an androgenous/heteronormative society – thus have “good reason” to have negative mood. I strongly suggest removing the biological deterministic views from the Introduction.
18) You need to cite a study that shows that the BRUMS is short and has robust psychometric properties (in what samples?).
RESPONSE: The suggested change in the Introduction has been implemented. Details of the original validation studies of the BRUMS are included in 2.1.1
>>> Further suggestion: What are “robust psychometric properties” on what sample – no information has been added to 2.1.1
19) First report in what language and for what sample the Scale was developed. Then, I suggest citing exemplars; however, it would be best to have a review available on the multiple uses of the scale.
RESPONSE: This information has been added on page 2.
>>> Further suggestion: This section has considerably improved in my opinion. However, I do not think “American mood descriptors” is a common term.
20) I do not think the geographical information on Republic of Lithuania (page 2) is necessary. It would be more relevant to know how mood differs on average and how mood has been conceptualized in Lithuania in comparison to the sample/population from which the BRUMS was developed from.
RESPONSE: This section has been removed as suggested.
>>> Further suggestion: I do not think the geographical information on Republic of Lithuania (page 2) is necessary. “In 2004, Lithuania became a member of the EU, and its language…”
Furthermore, what is an “EU language”?
21) Does the source [14] compare suicide rates in Lithuania with other EU countries? Why do you talk about suicide in specific?
RESPONSE: This section has been modified for clarity.
>>> Further suggestion: Why do you talk about suicide in specific? I am also not quite convinced that the BRUMS is an adequate “screening tool” in healthcare. Offer some studies that report about longitudinal findings that show the link between BRUMS scores and mental health problems.
What do you mean by “antecedents, correlates, and behavioral outcomes of moods in a Lithuanian context”?
22) You can give the explanation of the link between negative mood and mental health problem more space; then try to use models and explanations for this associations. Just picking some mental health problems seems arbitrary and confusing.
RESPONSE: The link between negative mood and mental health problems has been given more space on p.2.
>>> Further suggestion: You can give the explanation of the link between negative mood and mental health problem more space; then try to use models and explanations for this associations. Just picking some mental health problems seems arbitrary and confusing. No models have been discussed.
23) You can already in the Introduction explain, what scales we can expect; has mood dimensions/facets?
RESPONSE: This information has been added on p.2.
>>> Further suggestion: What do the words “Tension, Depression, Anger, Vigor, Fatigue, Confusion” mean? Why are there only those moods – what is the theoretical basis. Is happiness, content not also a possible mood state?
25) Try to already explain in the Introduction, what convergent/divergent validity you expect the scales to have.
RESPONSE: This information is provided in 2.1.3.
>>> Further suggestion: “convergent validity (i.e., whether subscale scores correlated highly with concurrent measures of similar constructs)” Which scales would that be? And why – what studies did find hypothesized associations? You mention it in 2.1.2; but it needs to be explained in more detail in the Introduction (it is a hypothesis that is tested; thus the rationale for the hypothesis needs to be clear in the Introduction).
For example why should extraversion be linked to negative mood (“because some subscales (e.g., Extraversion, Neuroticism) have been shown to correlate with negative mood scores”)?
26) Try to explain what steps are needed to translate the BRUMS (from English?) to Lithuanian. Use guidelines and suggestions such as 10.1016/j.bodyim.2018.08.014.
RESPONSE: Thank you for providing reference to these guidelines. I have added reference to them in the Discussion on p. 10.
>>> Further suggestion: Try to explain what steps are needed to translate the BRUMS.
Methods
27.1) Argue how and why your sample size was large enough – include some sample size, effect size, power analyses.
RESPONSE: An explanation is provided on p. 5.
>>> Further suggestion: You state that your “sample of 746 participants exceeded the recommended minimum sample size (i.e., 10 participants per item) for confirmatory factor analysis”; the N:q rule of thumb refers to number of model parameters (including error terms and correlations between scales), not items.
30) Discuss the POMS and the improvements made with the BRUMS (as you shortly do in 2.1.1.) already in detail in the Introduction.
RESPONSE: The POMS and the improvements associated with the BRUMS are now addressed in the Introduction on p. 2.
>>> Further suggestion: I do not think the relation between the POMS and the BRUMS is clear enough. What has been done to the POMS to get the BRUMS. What model is used? How can 24 items be answered to in 1 minute? Give an example that the items consist of only one word each.
“because the TMD score treats Tension and Anger as inherently negative mood” – did you not treat “negative mood” to be inherently negative and associated with mental disorders, too? Explain in the Introduction also positive consequences of “negative mood”.
41) Was data assessment after lockdown? “This period represented the end of the COVID-19 pandemic second wave, a lengthy low in reported cases, and then a sharp return of a third wave” was the “sharp return” still during data assessment. This part does not seem clear to me.
RESPONSE: This section has been rewritten to clarify the situation (see 2.2).
>>> Further suggestion: I do not think Figure 1 is necessary.
41) Reword “Good-fitting models tend to produce consistent results on different measures of model fit” rather maybe state “the model needs to fit the data, indicated by consistent results on different measures of model fit”
RESPONSE: The suggested change has been implemented.
>>> Further suggestion: “However, the χ2 value is always significant for samples with ≥ 400 cases, and hence it was considered an unsuitable statistic for our purpose” Still it is recommended to report Chi.
Results
50) How can the correlation of error term between items that do not load on the same scale be justified?
RESPONSE: The pathway in question was between the Anger and Tension subscales, not item scores. We have rewritten this sentence in 3.2 to clarify.
>>> Further suggestion: “and can be defended theoretically [72]” What is the explanation? However, move the interpretation of results to the Discussion.
51) “All items loaded on target factors with loadings above .50” report all loadings, e.g., in a table.
RESPONSE: Item loadings are now reported in Table 3.
>>> Further suggestion: Why are factor loadings for Worried, Panicky, Unhappy not reported? Have those items significant loadings on Tension?
To which scale do “Bad tempered, Bitter, Angry” belong to?
Why do many items (e.g., Tired, Worn out) have high loadings on two scales?
53) “The opposite was true for the Vigor subscale”, namely (spell it out)?
RESPONSE: Results for Vigor have been added to 3.4.
>>> Further suggestion: What are “negligible inverse correlations”? Not significant? Or low effect size? Was the correlation in Table 5 with r = -.07 significant?
In women some correlations between age and the mood scales reached |r| > .2 Why do you interpret the correlations of the Big Five with the mood scales differently than the correlation between age and the mood scales?
54) “Between-group MANOVAs were conducted to determine if mood scores varied by sex” use “gender”
RESPONSE: The term sex has been replaced by gender throughout the manuscript.
>>> Further suggestion: also change the label “F” to Women and “M” to “Men” in Table 6
If you include moth of assessment/participation in your analysis (3.7.) you also need to report your sample segregated by month.
“The observed lack of significant differences in mood scores during different phases of the pandemic suggests that mood did not fluctuate greatly during our data collection period.” – or your power was to low.
57) Do an invariance analysis across gender to see whether the scales measures the same in women and men.
RESPONSE: An invariance analysis has been conducted and is reported in Table 2 and 3.2
>>> Further suggestion: No invariance test has been conducted (see Kline chapter 16).
Discussion
62) It does not seem clear what you mean by “concurrent” – is this convergent or divergent validity?
RESPONSE: This has been explained in 3.4.
>>> Further suggestion: Use more common terms such “convergent or divergent validity” – mixing those terms does not seem to increase clarity.
64) “generation of a table of normative scores for the LTUMS” I think this is a major limitation of your study, to not have assessed norm values. It does not seem correct to compare “raw scores” with normative scores from another population (and measure).
RESPONSE: We have clarified that the table of normative scores in Table 3 is derived from the Lithuanian sample (Discussion, 1st para.) and no comparisons with previous normative datasets are now included.
>>> Further suggestion: Table 3 are factor loadings.
What “norm population” do you use? You cannot use the same “norm population” you do the rest of the analysis. Furthermore, a more detailed description of the sample would be needed, e.g., psychological parameters.
You later report that women and men have different mean scores – thus, different “norms” seem to be needed for women and men. 199 women do not seem enough to define a “norm population” for the scale.
65) What are “mood profiles”, and why are they important?
RESPONSE: Mood profiles and their importance are now explained (Introduction, p.2; Discussion, 2nd para.)
>>> Further suggestion: Should have been explained in the Introduction.
66) Compare your psychometric results to other validation studies – what can be improved? Why do error terms need to correlate?
RESPONSE: addressed, pg. 10.
>>> Further suggestion: I do not see comparisons of this scales psychometric properties with the original scale on page 10. Furthermore, it needs to be discussed why certain error terms need to correlate; and also why certain items do not load on their scale; as well as why certain items seem to load on more than one scale – according to Table 5.
68) Do not rely solely on biological concepts for explaining gender differences in mood. Being subject to discrimination and gender disparities that disadvantage women to a higher degree might very easily explain negative mood in woman compared to men – this has nothing to do with the “inability to effectively downregulate negative feelings”
RESPONSE: We apologise for this simplistic approach and have now duly recognized societal explanations for mood differences by gender in the Introduction on p.2.
>>> Further suggestion: I do not think changes go far enough. The explanation is still biologically deterministic.
70) Offer some more reflection on the Limitations, e.g., how has the questionnaire been translated? Did you follow all suggestions by 10.1016/j.bodyim.2018.08.014?
RESPONSE: An additional section has been added to the Discussion section to address this issue.
>>> Further suggestion: The offered source states that “translation-back translation” is the bare minimum, and should be improved, by testing cultural appropriateness of the scale. It seems very difficult, to translate one word statements into a different language.
75) Discuss the results of convergent and divergent validity in more detail – also what they mean.
RESPONSE: Convergent and divergent validity have been introduced on p. 3, mentioned in the Methods 2.1.2 and 2.1.3, and in the Results 3.4.
>>> Further suggestion: You need to include them in your Discussion as well. They are part of your hypothesis. Also they signify how mood is linked to other psychological variables – thus might offer some “meaning”.
Also you never mention psychological health problems in the discussion again. Considering how much you focused in the Introduction on “metal health problems” the Introduction and Discussion do not seem to complement each other.
Author Response
Please see attachment. It is the same response as it was before our third revision. Thank you.

Round 3
Reviewer 2 Report
Thank you for letting me read your manuscript of “Validation of a Lithuanian-language version of the Brunel Mood Scale: the BRUMS-LTU”
I commend the authors for making considerable changes to their manuscript! I think that the manuscript has substantially improved!
Minor changes recommended:
Avoid using the word “normal” (page 3)
Table 2. explain “modified” in the table’s notes
Formating of Table 3 seems to need corrections
“negligible-to-very weak (r ≤ .20)” – use “small effect”
Correct the first sentence of the Discussion. The sentence does not seem finished.
Author Response
Please see attached response. Thank you.
